# Targeting Replication Stress Using CHK1 Inhibitor Promotes Innate and NKT Cell Immune Responses and Tumour Regression

**DOI:** 10.3390/cancers13153733

**Published:** 2021-07-25

**Authors:** Martina Proctor, Jazmina L. Gonzalez Cruz, Sheena M. Daignault-Mill, Margaret Veitch, Bijun Zeng, Anna Ehmann, Muhammed Sabdia, Cameron Snell, Colm Keane, Riccardo Dolcetti, Nikolas K. Haass, James W. Wells, Brian Gabrielli

**Affiliations:** 1Mater Research Institute, The University of Queensland, Brisbane, QLD 4102, Australia; martina.proctor@mater.uq.edu.au (M.P.); ehmann.anna@gmail.com (A.E.); m.bilalsabdia@mater.uq.edu.au (M.S.); colm.keane@mater.uq.edu.au (C.K.); 2The University of Queensland Diamantina Institute, The University of Queensland, Brisbane, QLD 4102, Australia; j.gonzalezcruz@uq.edu.au (J.L.G.C.); s.daignault1@uq.edu.au (S.M.D.-M.); m.veitch@uq.edu.au (M.V.); b.zeng1@uq.edu.au (B.Z.); Riccardo.Dolcetti@petermac.org (R.D.); n.haass1@uq.edu.au (N.K.H.); j.wells3@uq.edu.au (J.W.W.); 3Mater Pathology, Mater Research, Mater Hospital, Raymond Terrace, South Brisbane, QLD 4101, Australia; cameron.snell@mater.org.au; 4Peter MacCallum Cancer Centre, Melbourne, VIC 3000, Australia; 5Sir Peter MacCallum Department of Oncology, The University of Melbourne, Melbourne, VIC 3010, Australia; 6Department of Microbiology and Immunology, The University of Melbourne, Melbourne, VIC 3010, Australia

**Keywords:** melanoma, CHK1 inhibitor, replication stress, immunogenic cell death, innate immune response, adaptive immune response, NKT cells

## Abstract

**Simple Summary:**

Therapies selectively targeting cancer-specific defects have the advantage of minimising damage to normal tissue including the immune system. The work described here investigates a therapy that targets replication stress, a common feature of many cancer types including melanoma. We demonstrate that this therapy not only selectively kills tumours but also triggers pro-immunogenic signals from the tumour to attract immune cells. In animal models, it has no adverse effects on immune response and triggers a strong anti-tumour immune response. The major component of this response are specialised immune cells, but the tumour itself trigger a conversion of this anti-tumour response to an immune suppressive response that cannot be overcome with current immunotherapies. The work demonstrates that understanding the immune response triggered is essential to guide the selection of the optimal immunotherapy to promote long term tumour control.

**Abstract:**

Drugs selectively targeting replication stress have demonstrated significant preclinical activity, but this has not yet translated into an effective clinical treatment. Here we report that targeting increased replication stress with a combination of Checkpoint kinase 1 inhibitor (CHK1i) with a subclinical dose of hydroxyurea targets also promotes pro-inflammatory cytokine/chemokine expression that is independent of cGAS-STING pathway activation and immunogenic cell death in human and murine melanoma cells. In vivo, this drug combination induces tumour regression which is dependent on an adaptive immune response. It increases cytotoxic CD8^+^ T cell activity, but the major adaptive immune response is a pronounced NKT cell tumour infiltration. Treatment also promotes an immunosuppressive tumour microenvironment through CD4^+^ Treg and FoxP3^+^ NKT cells. The number of these accumulated during treatment, the increase in FoxP3^+^ NKT cells numbers correlates with the decrease in activated NKT cells, suggesting they are a consequence of the conversion of effector to suppressive NKT cells. Whereas tumour infiltrating CD8^+^ T cell PD-1 and tumour PD-L1 expression was increased with treatment, peripheral CD4^+^ and CD8^+^ T cells retained strong anti-tumour activity. Despite increased CD8^+^ T cell PD-1, combination with anti-PD-1 did not improve response, indicating that immunosuppression from Tregs and FoxP3^+^ NKT cells are major contributors to the immunosuppressive tumour microenvironment. This demonstrates that therapies targeting replication stress can be well tolerated, not adversely affect immune responses, and trigger an effective anti-tumour immune response.

## 1. Introduction

The clinical deployment of immune checkpoint inhibitors (ICI) has had a major impact on the treatment of melanoma, particularly with respect to improvements in long term survival [1]. However, ICI is only effective in immunologically “hot” tumours that already have significant immune cell infiltration, and not in immunologically “cold” tumours that lack immune cell infiltration [2]. In 50% of patients with mutant BRAF, BRAF/MEK inhibitors can modify the tumour immune microenvironment to enhance ICI efficacy [3]. Unfortunately, no such options currently exist for BRAF wild type patients, for whom 5-year survival on ICI is <40% [4].

One approach to increasing the proportion of responders and duration of the response is to enhance the immunogenicity of the tumours. This can be achieved by selectively killing tumours in a manner that stimulates an immune response. An example of this approach is the combination of ICI and stereotactic radiation which improves overall survival [5]. This immune response is mediated by increased tumour antigen presentation and release of danger-associated molecular patterns (DAMPs) from radiation-induced dying tumour cells [6]. DAMPs are presented in the form of released or surface exposure of intracellular components such as HMGB1, ATP, heat shock proteins HSP70 and HSP90, and Calreticulin. The release or relocation of DAMPs mark cells undergoing immunogenic cell death (ICD), and in turn attract and activate effector immune recognition and responses [7]. ICD also requires tumour cell production of pro-inflammatory cytokines Type I interferons (IFN), IL-1β, TNFα and CXCL10, which recruit and activate antigen-presenting cells such as dendritic cells (DCs), natural killer cells (NK, NKT cells) and macrophages, and enhance MHC class I expression [8].

ICD can be promoted by some conventional chemotherapeutic agents [7,9], but a shortcoming of these drugs is they are commonly strongly myelosuppressive, effectively blocking the desired immune responses [10]. Checkpoint kinase 1 inhibitors (CHK1i) as single agents have shown preclinical activity although only limited efficacy in a clinical trial [11,12,13]. CHK1i selectively target cells with elevated levels of endogenous or exogenously generated replication stress [11,14]. CHK1i effectively combine with gemcitabine and have shown clinical activity, although patients commonly suffered severe adverse haematological events [15,16,17]. We have previously reported that subclinical doses of CHK1i strongly synergise with subclinical doses of the replication inhibitor hydroxyurea (low dose; LDHU) to block tumour growth in a high proportion of melanomas and NSCLC [14,18]. In vivo, the effect is long-lasting, and is associated with recruitment of macrophages in immune-compromised mice, with little effect on normal tissue proliferation, including myeloid cells.

CHK1i+LDHU synergise to promote high levels of DNA damage, the result of excessive replication origin firing, and cell death [18]. Increased DNA damage can trigger activation of the cGAS-STING pathway responsible for increased pro-inflammatory cytokine and chemokine expression that can recruit lymphoid and myeloid cells to control tumour growth [19]. Inhibiting the DNA damage response to radiation by inhibiting ATR, the activator of CHK1 triggers a durable immune response associated with a reduced presence and activity of immunosuppressive regulatory T cells (Treg) and reduced levels of PD-1 on activated T cells [19].

The approach investigated here is to exploit replication stress, a vulnerability in many cancer types including melanomas [11,14,20]. We demonstrate that CHK1i+LDHU triggers ICD, increases immune chemoattractant cytokine and chemokine expression, and activates an anti-tumour immune response. This delivers the two-fold benefit of immediate tumour growth control coupled with the promotion of anti-tumour immune responses.

## 2. Materials and Methods

### 2.1. Melanoma Cell Lines

Six human melanoma cell lines used in this study were cultured as 2D attached cell lines or as tumourspheres as described previously [11]. The mouse melanoma cell lines YUMM1.7 and YUMM3.3 were used in this study [21]. YUMM1.7 and YUMM3.3 were irradiated with 300 Jm^−2^ UVB radiation in a BioSun (Vilber Lourmat; Torcy, France) UV irradiator (peak emission 312 nm), three times, with sufficient time between irradiation to allow recovery (4–6 days), to generate the YUMMUV1.7 and YUMMUV3.3 lines. YUMMUV1.7 and YUMMUV.3 were then immunoedited by growing as subcutaneous tumours in immunocompetent C57BL/6J mice to >300 mm^3^ then harvested. Tumours were cut into 2 mm cubes, minced into a slurry, and pushed through a 70 mm cell strainer. Recovered cells were washed and seeded into a flask with 10% FBS/RPMI1640 growth media supplemented with antibiotics. After one day, any unattached cells were removed and replaced with fresh growth media, then cultured until confluent and passaged. Immunoediting was performed once for YUMMUV1.7 and twice for the YUMMUV3.3 line, and these lines were used for the subsequent experiments. The cell lines tested negative for Mycoplasma and appeared morphologically identical to the parent lines. The viability of YUMMUV1.7 and YUMMUV3.3 cells was assessed using resazurin assay as previously described [22].

### 2.2. Immunoblotting

Frozen cell pellets where lysed and supernatants prepared and immunoblots prepared as described previously [11]. Membranes were probed with antibodies to detect IRF-3(D83B9;Cat# 4302), phospho-IRF-3 (Ser396; Cat#29047), STING (D2P2F; Cat#13647), phospho-STING (Ser366; Cat#19781), TBK1 (D1B4; Cat#3504), phospho-TBK1 (Ser 172; Cat# 5483), NF-kappa B p65(D14E12; Cat# 8242), phospho-NF-kappa B p65 (Ser536; Cat# 3033), cleaved PARP1(Asp214; Cat#9541), phospho-Histone H2A.X (Ser139; Cat#2577) from Cell Signalling (Arundel, QLD, Australia) and PD-L1 (Roche, Sydney, NSW Australia).

### 2.3. Mouse Studies

All animal experiments were approved by the University of Queensland Animal Ethics Committee using approval number MRI-UQ/211/17. Four to six-week-old female C57BL/6J and C57BL/6J *Rag1^−/−^* mice were obtained from the Animal Resources Centre (Perth, WA, Australia). Mice were injected with 0.5–2 × 10^6^ tumour cells in matrigel (Cultrex, Trevigen, Gaithersburg, MD, USA) subcutaneously in the shaved hind flank. Once tumours reached 100–200 mm^3^, mice were treated with either vehicle (10% DMSO, 5% Tween80, 5% PEG400 in clinical-grade saline) or 50 mg/kg SRA737 (provided Sierra Oncology, San Mateo, CA, USA) combined with 100 mg/kg HU in the vehicle by oral gavage, then 4 h later by i.p. injection of 50 mg/kg HU (in saline) for three cycles where one cycle is treatment on three alternative days a week. Tumour size was measured three times per week using callipers. Mice were treated with anti-mouse PD-1 mAb (RMP1-14), CTLA4 mAb (9H10) or rat IgG2A isotype control (2A3) mAb (Bio X Cell, Lebanon, NH, USA) at 200 mg/mouse initially on day 8 following the first dose of CHK1i+LDHU treatment then two days later. This was repeated for two more weeks at two days apart during the drug treatment and for one further week following completion of drug treatment.

### 2.4. Cytoplasmic DNA Immunofluorescence

Melanoma cell lines were plated in 96 well plates and treated with CHK1i+LDHU for 48 h, then fixed using 4% PFA and washed in PBS. Cells were stained with anti-dsDNA Ab (Merck) and developed with the appropriate fluorescently labelled secondary antibody. Plates were imaged in an InCell Analyzer 2000 (Cytiva, North Ryde, NSW, Australia) and the cytoplasmic pixel intensity analysed as previously [23]. To validate the specificity of the dsDNA Ab staining, fixed cells were treated with DNAse1 (Roche) following the permeabilisation step by adding 200 U per well and incubating at 37 °C for 30 min before washing in PBS and proceeding with the blocking step. A secondary antibody alone control was also used.

### 2.5. Immunohistochemistry

Paraffin-embedded tumour sections were stained with rat anti-mouse CD8α Ab (1:200; eBioScience, Waltham, MA, USA) following antigen retrieval in 10 mM sodium citrate buffer pH 6.0 and detected with anti-rat IgG-HRP polymer kit (Vector Labs) using Nova Red as a substrate (Vector Labs). PDL-1 staining of paraffin-embedded tumour sections was carried out by Mater Pathology (Mater Hospital, Brisbane, Australia) according to the manufacturer’s instructions using the SP142-PDL-1 Roche/Ventana Medical Systems reagents.

### 2.6. NanoString Analysis

YUMMUV cell line tumours were harvested from mice at the completion of treatment with CHK1i +LDHU, cut in half and were snap-frozen. Total RNA extracted using a miRNEasy Kit (Qiagen, Chadstone, VIC, Australia) from frozen tumour tissue. A total of 150 mg RNA from each tumour was used in the NanoString panel using the Mouse Pan-Cancer Immune Profiling Panel according to the manufacturer’s protocol. Data were analysed using the NanoString nSolver program.

### 2.7. Flow Cytometry

#### 2.7.1. ICD Markers

Human melanoma cell lines were cultured for 48 h in the presence of DMSO or CHK1i +LDHU, then harvested and stained for surface expression with anti-CRT-APC, anti-CD47-APC, anti-HSP70-FITC, anti-HSP90-PE or Relevant isotype controls. Mean fluorescence intensity (MFI) values were calculated by taking the geometric mean values. Stained cells were analysed using an LSR-Fortessa X20 Flow Cytometer (BD BioSciences) with FACSDiva^TM^ software (Becton Dickinson, Sparks, MD, USA). Acquired data were analysed using FlowJo software (TreeStar Inc., Ashland, OR, USA).

#### 2.7.2. Immune Profiling

Tumours were harvested, minced and treated with DNASe1 and collagenase IV for 30 min, then pushed through a 40 mM cell strainer to generate a single-cell suspension. Cells were blocked using Fc block then stained with Live/dead Aqua (Thermo Fisher Scientific, Waltham, MA, USA) and a panel of conjugated T-cell (CD45.2-PE dazzle, TCRβ-FITC, CD8α-BV605, CD4-AF700, PD-1-BV711, NK1.1-PE-Cy7) or myeloid antibodies (CD45.2-PE-dazzle, F4/80-FITC, CD11c-PE, MHCII-APC-Cy7, CD11b-BV650, Ly6C-APC, Ly6G-PE-Cy7). FoxP3 was detected using the FoxP3 Staining Kit (eBioscience) as per the manufacturer’s instructions. Flow-Count Fluorospheres (Beckman Coulter, Miami, FL, USA) were used for total cell counts. Single Ab stained compensation beads were used to set the gating. Stained cells were analysed using an LSR-Fortessa X20 Flow Cytometer (BD BioSciences) with FACSDiva software. Acquired data were analysed using FlowJo software.

#### 2.7.3. Polyfunctional T Cell Assay

Blood was collected from mice into EDTA treated tubes and treated with ACK (Ammonium–Chloride–Potassium) buffer treated to lyse erythrocytes. The functional activity of CD8^+^ and CD4^+^ T cells in blood were quantified using intracellular FACS staining for IFN-g, TNF-a, and IL-2 after stimulation with the combination of survivin (SUR_53-67;_ MHCI and MHCII epitopes) and tyrosinase-related protein-2 (TRP2_180-188_ MHCI epitope) peptides in the presence of Brefeldin A for 5 h. Cells were surface stained with anti-CD3e-AF488, anti-CD4-PE-Cy7, anti-CD8-APC/Cy7, then intracellularly stained with anti-TNFα-AF647, anti-IFNγ-AF700, anti-IL-2-PE following fixation and permeabilising. Samples were analysed using an LSR-FortessaX20 Flow Cytometer. Cytokine co-expression profiles were quantified using the Boolean gating function of Kaluza software.

#### 2.7.4. Cytokine Bead Array

LegendPlex Human anti-Virus Response Panel (BioLegend Cat #740390) was used according to the manufacturer’s instructions to assess the cytokines present in cell supernatants after 48 h treatment. Samples were performed in triplicate and analysed on a CytoFLEX S Flow Cytometer (Beckman Coulter, Lane Cove, Australia). Acquired data were analysed using provided LEGENDplex^TM^ Data Analysis Software (BioLegend, San Diego, CA, USA).

### 2.8. ELISpot Assay of Mouse Splenocytes

HPV16-E7-specific CD8^+^ T-cell responses were measured using IFNγ ELISpot as described previously [24].

## 3. Results

### 3.1. CHK1i+LDHU Triggers Pro-Inflammatory Cytokine Expression Independent of cGAS-STING and Immunogenic Cell Death

We have previously shown that subclinical doses of CHK1i and LDHU are ineffective as single agents but synergise strongly to control tumour growth of human melanoma and NSCLC cells in vitro and xenografts in immune-compromised mice [18]. On the basis of our previous studies, we used the CHK1i+LDHU combination only for these studies. Combination treatment triggered cell death in all lines (Appendix A), and the loss of viability was reversed using 100 mM of the pan-caspase inhibitor zVAD (Appendix A). We previously observed that CHK1i+LDHU treatment increased macrophage infiltration and HMGB1 relocation from the nucleus [18], suggesting that other DAMPs may also be externalised. Increased surface expression of DAMPs calreticulin (CRT), HSP70 and HSP90 was also detected in a panel of melanoma tumoursphere lines (Figure 1A), and CD47, the inhibitory signal for phagocytosis by immune cells, was increased in three of the six lines tested (Appendix A). Levels of pro-inflammatory cytokines and chemokines including CXCL10, IFNα, IFNβ, IL-1b and TNFα increased with treatment in the majority of the melanoma tumoursphere lines tested (Figure 1B). CXCL10 expression was shown to be increased by activation of the cGAS-STING pathway in response to increased cytoplasmic DNA to activate TBK1 and IRF3 [25,26]. CHK1i+LDHU treatment increased DNA damage in all sensitive tumoursphere lines, indicated by increased cleaved PARP, and this was correlated with increased cytoplasmic double-stranded DNA (Figure 1C,D). However, no evidence of cGAS-STING-TBK1-IRF3 pathway activation was observed with treatment. cGAS and STING were absent or expressed at very low levels in four of the six lines assessed, and there was no evidence of activation with either 24 h (Figure 1D) or 48 h treatment (not shown). There was an increase in the level of PD-L1 with drug treatment, although its level varied greatly from cell line to cell line (Figure 1D).

These data indicated that CHK1i+LDHU treatment triggered ICD in human melanomas, although the increased PD-L1 expression may dampen the immune response. To validate the ICD triggered by treatment, we tested our drug combination on two mouse melanoma lines (YUMMUV1.7 and YUMMUV3.3) derived from YUMM1.7 (Braf V600E:Cdkn2a^−/−^:Pten^−/−^) and YUMM 3.3 (Braf V600E:Cdkn2a^−/−^) [21] and subjected to ultraviolet radiation in vitro utilising a modified protocol described [27]. These UV irradiated cells were also immunoedited by serial passage through immunocompetent mice to produce cell lines that were comparable to human tumours that are naturally immunoedited during their development [28]. Both cell lines were very sensitive to the CHK1i+LDHU combination in vitro (Appendix A). To determine the immunogenic potential of the CHK1i+LDHU in vivo, YUMMUV1.7 and YUMMUV3.3 cells treated in vitro with CHK1i+LDHU for 48 h were used to inoculate immunocompetent mice. After 12 days, mice were challenged with live tumour cells injected into the opposite flank and tumour growth was followed. Freeze-thawed tumour cells were used as a negative control and doxorubicin treated cells as a positive control for ICD. Tumour grew at similar rates in the control and freeze-thawed cell inoculated mice, but 5/5 of doxorubicin and 3/5 CHK1i combination-treated YUMMUV1.7 inoculated mice were completely protected, tumour growth in the remaining two mice was significantly retarded (Figure 2A,B). Tumour growth was detected in 1/5 of the mice immunised with CHK1i+LDHU-treated YUMMUV3.3 (Appendix A).

The tumour-free mice were rechallenged with live tumour cells eight weeks after initial inoculation and no tumour outgrowth at either the initial inoculation or challenge sites was found even at >80 days. When all of the control tumours had reached >1 cm^3^ on day 27, small tumours (<100 mm^3^) were detected in 1/5 doxorubicin and 2/3 CHK1i combination rechallenge mice (Figure 2C; Appendix A). Tumour protection appeared to be linked to the T cell compartment as there was a significant increase in tumour-specific IFN-γ production in splenocytes from CHK1i combination-treated mice (Figure 2D). The protective effect was lost in Rag1^−/−^ mice that lack adaptive immune cells, indicating the requirement for an adaptive immune response in the protection tumour growth (Figure 2E).

### 3.2. CHK1i+LDHU Does Not Adversely Affect T-Cell Proliferation or a T-Cell Mediated Immune Response

Many cytotoxic drugs that stimulate ICD and cytokine expression are also toxic to proliferating immune cells, effectively suppressing an immune response. To investigate the effect of CHK1i+LDHU on CD8^+^ T cell activation, the ability of mice to mount an effective immune response to a model antigen, HPV-16 E7 [29] was assessed. Both treated and untreated mice vaccinated with E7 peptide effectively prevented TC-1 tumour growth (Figure 3A), and ELISpot assay of spleen derived T cells revealed strong T cell activation to E7 peptide (Figure 3B). Three-day treatment with CHK1i+LDHU also had little effect on the proliferation and viability of human T cells. By contrast, treatment with the Aurora kinase inhibitor Alisertib which is known to promote leukopenia [30], reduced T cell proliferation. (Figure 3C) These data demonstrate that CHK1i+LDHU does not adversely affect T cell-mediated immune responses.

### 3.3. CHK1i+LDHU Triggers an Active Immune Response In Vivo

To investigate the immune response triggered by in vivo administration of CHK1i+LDHU, syngeneic YUMMUV1.7 and YUMMUV3.3 melanomas were established as subcutaneous tumours in immune-competent C57BL/6J mice. Treatment with SRA-737+LDHU in vivo effectively inhibited tumour growth (Figure 4A; Appendix A). The immune response within treated and untreated (control) tumours was initially assessed by transcriptomic analysis using the NanoString Mouse PanCancer Immune Profiling panel of 770 immune-related genes. NanoString defined signatures for immune cell types indicated significant changes in dendritic and NK cells, Th1, cytotoxic and exhausted T cells, Tregs and macrophages (Appendix A). Using a curated gene list revealed that the two models increased markers of T cell activation (Eomes, Txk) and cytotoxic activity (Fas, Gzma/b and Prf1) but also markers of immune suppression (CTLA4, FoxP3, PD-L1), although the extent of changes was generally more modest in the YUMMUV1.7 model (Figure 4B,C), a consequence of the increased immunosuppression promoted by Pten deletion in the latter model [31]. Although expression analysis showed only small changes in CD8^+^ T cell markers, immunohistochemical staining of the tumours for CD8a showed increased numbers of infiltrating CD8^+^ cells with drug treatment (Appendix A). Both models showed increased expression of NK cell markers (Klrg1, Klra7, Klrb1c/NK1.1, Ncr1; Figure 4D). Dendritic cells appeared to be cDC1 type (CD103, Xcr1, CD11c, Clec4n) with a greater increase in the YUMMUV3.3 model (Figure 4E).

Similar to the parental lines, [21] both YUMMUV tumours contain >30% F4/80^+^ macrophages. Drug treatment increased markers of classical activation (iNOS, Msr1; M1) and decreased alternative activation markers (Mcr1/CD206, CD163; M2) in both models (Figure 4F). MAST cells detected by c-Kit/CD117expression were unchanged with treatment (data not shown). There was also increased expression of a number of complement system components in both models (Appendix A).

Increased expression of MHC Class I was found in both models following drug treatment, whereas MHC Class II gene expression was consistently increased in YUMMUV3.3 but decreased in YUMMUV1.7 (Figure 4G). Expression of antigen processing components was also increased in both treated groups. The expression of monocyte/macrophage chemoattractant Ccl2/MCP-1, Ccl7, Ccl8/MCP-2 [32,33], and leukocyte chemoattractant Ccl21, Ccl5/Rantes and Cxcl10 [34,35,36], were increased. Interestingly, there was a down-regulation of pro-metastatic and angiogenic factor Cxcl12 and its receptor Cxcr4 [37] in both models (Figure 4H). Isg20 and Zbp-1 are IFN-induced anti-viral response genes that trigger NF-kB activation and cytokine expression [38,39] were upregulated with treatment. Ccl2, Ccl5, Cxcl10, Cxcl12, Cxcr4, Isg20 and Stat1 are RelA/NF-kB regulated genes, and increased pRelA was found in drug-treated YUMMUV1.7 cells consistent with increased DNA damage (Appendix A), whereas no evidence of STING pathway activation was found (Appendix A).

The immune gene expression profile guided the selection of a 14-marker panel separated into myeloid and lymphoid sets that were used to investigate changes in tumour infiltrating CD45^+^ cells using flow cytometry. The treated YUMMUV1.7 tumours were sampled at two-time points, day 10 after treatment commenced (day 27 after tumour injection), and day 24, after the last treatment. The controls were harvested on day 10. Unsupervised clustering of the CD45^+^ cells was performed to identify the major immune cell types and the changes in absolute numbers relative to tumour was calculated. In the myeloid compartment, F4/80^+^ monocyte/macrophages were the most abundant cell types, present as three major subtypes, F4/80^+^only, and two populations of F4/80^+^ CD11b^+^ MHCII^hi^; the Ly6C^+^ subset paralleling the F4/80^+^ only population, peaking at day 10 then reducing, whereas the Ly6C^−^ subset accumulated only at day 24 (Figure 5A,B). The CD11b^+^ neutrophil population also increased significantly at day 10 to return to control levels by day 24. Similar subsets and trends were observed in the YUMMUV3.3 model (Appendix A).

The lymphoid cell clusters revealed altered levels of CD8+ T cells, NK and NKT cells, and immunosuppressive CD4^+^ Treg and FoxP3^+^ CD25^+^ NKT cells (Figure 5C). There was a modest increase in CD8^+^ T cells, and the majority of these were PD-1 positive (Figure 5D). Further analysis of the flow profiles using standard gating approaches demonstrated that in 2/4 controls, approximately 30% of CD8α^+^, TRCβ^hi^, NK1.1^−^ T cells were positive for PD-1, the other two were similar to the treated tumours and >90% strongly PD-1 positive (Figure 5E). The high-level PD-1 expressing controls were the larger tumours (~1 cm^3^). PD-1 expression was restricted to CD8^+^ T cells. There was a modest increase in NK cells and a prominent increase in Tregs by day 24. The most striking feature was the accumulation of NKT cells and FoxP3^+^ CD25^+^ CD69^−^ NKT cells at day 10 which then decreased concomitantly with an increase in FoxP3^+^ CD25^+^ CD69^+^ suppressive NKTs [40] at day 24 (Figure 5E). This suggested that the immunosuppressive population was derived from the infiltrating NKT cells while resident in the tumour. The absolute numbers of these suppressive cell types exceeded the number of potential effector T and NKT cell populations. A similar outcome was observed in the YUMMUV3.3 model, although the increase in CD8^+^ T cells was more apparent Appendix A). The immune suppression appeared limited to the tumour microenvironment as CD4^+^ and CD8^+^ T cells extracted from the peripheral blood of both models strongly reacted ex vivo upon stimulation with tumour antigens (Figure 5F, Appendix A). Together these data indicate that the CHK1i+LDHU combination promotes strong anti-tumour immune responses, but this appeared to be blunted in the tumour microenvironment by T cell exhaustion and immunosuppressive signals from Tregs and FoxP3^+^NKT cells.

The contribution of the adaptive immune response to tumour control was investigated in Rag1^−/−^ mice using the more aggressive YUMMUV1.7 model. Untreated tumours grew rapidly in both Rag1^−/−^ and C57BL/6 mice, however, CHK1i+LDHU treated C57BL/6 controlled tumour growth more effectively than Rag1^−/−^ mice, indicating a role for the adaptive immune system (Figure 6A; Appendix A).

### 3.4. Efficacy of the CHK1i+LDHU Combined with Immune Checkpoint Blockade

The high level of PD-1 expression in CD8^+^ T cells suggested T cell activation followed by exhaustion. To investigate whether inhibiting this immunosuppressive pathway might prolong CD8^+^ T cell activity in the tumour microenvironment, anti-PD-1 treatment was commenced on day 8, before the appearance of PD-1^+^ CD8^+^ T cells (day 10) in the YUMMUV1.7 model. No increased benefit was observed when PD-1 antibody treatment in combination with CHK1i+LDHU was compared to CHK1i+LDHU in combination with an isotype control antibody (Figure 6B, Appendix A). This was not due to the lack of effect of the PD-1 blockade as peripheral CD8^+^ T cells from CHK1i+LDHU and PD-1 antibody-treated mice had increased ex vivo stimulated activity compared to CHK1i+LDHU and isotype treated mice (Appendix A). A similar outcome was observed in the YUMMUV3.3 model. In this slower-growing model, the effect of anti-CTLA4 to target the Treg cells was also investigated by treating the remaining PD-1 isotype-treated mice. No significant difference between the groups was observed (Figure 6C), although there was a consistent long-term tumour suppression in 20% of CHK1i+LDHU-treated mice (Appendix A).

The level of PD-L1 was found to increase dramatically in tumour cells of mice treated with CHK1i+LDHU (Appendix A). The controls generally showed low-level staining across the tumour, although much of this appeared to be on smaller cells likely to be an immune cell population. The treated tumours all showed increased PD-L1 staining (Appendix A).

## 4. Discussion

Here we demonstrated that CHK1i+LDHU treatment of human and murine melanomas triggers ICD and pro-inflammatory cytokine and chemokines expression that promotes an adaptive anti-tumour cytotoxic and memory response. The ability to promote ICD and cytokine expression was shared by the two CHK1i used in this study, GDC-575 and SRA737 indicating the effects is a consequence of CHK1 inhibition and not an off-target effect of SRA737. The ability of CHK1i+LDHU to trigger cell death was related to the level of DNA damage induced by the treatment [18]. The ability to upregulate ICD markers such as relocation of CRT, HSP70 and HSP90 to the cell surface varied between cell lines, with only one of six human melanoma lines failing to show increased marker relocation. Interestingly, increased ICD markers and cytokine/chemokine levels did not always correspond with changes in DNA damage and cell death. The in vivo ICD assay demonstrated a strong adaptive immune response triggered by CHK1i+LDHU in vitro treated melanomas. The ability of the animals to mount a CD8^+^ T cell-dependent immune response in the presence of CHK1i+LDHU drugs and T cells to proliferate after treatment, in addition to our previous work showing that the drug combination had minimal effects on normal tissues [18], indicate that CHK1+LDHU treatment is compatible with therapies that engage the adaptive immune system.

DNA damaging agents were shown to promote pro-inflammatory cytokine/chemokine expression through a number of mechanisms including NF-κB and cytoplasmic DNA sensors such as cGAS-STING [41,42,43]. Sen et al. [25] have previously reported that a similar CHK1i combination robustly activated STING-dependent cytokine/chemokine expression and was critical for immune responses in small cell lung cancer models. However, dysregulation of cGAS-STING observed here in both human and mouse melanomas appears to be a common feature of melanomas that contributes to the immune evasion of these tumours [44]. Furthermore, it has recently been reported that CHK1i does not activate cGAS-STING signalling even in cells where this pathway is functional, and actually inhibits cGAS-STING pathway activation of IRF3 [45]. The ability of CHK1i+LDHU to promote strong pro-inflammatory cytokine/chemokine expression independent of this pathway suggests other pathways such as NF-κB observed in mouse melanoma. Other cytoplasmic DNA sensors such as ZPB1 can activate both NF-κB and inflammasome signalling, and cell death [39], and is strongly upregulated by CKH1i+LDHU may also have a role. It is unclear why melanomas and small cell lung cancer utilise different responses to promote similar pro-inflammatory responses. A possible explanation is that cGAS-STING is the primary mechanism for detecting and signalling in response to cytoplasmic DNA; loss of this pathway allows other default pathways to trigger a similar response.

The contribution of the adaptive immune response to tumour control by CHK1i+LDHU treatment is demonstrated by the inability of treatment to control tumour growth in the *Rag^−/−^* mice. The treatment was shown to increase intra-tumour infiltration of CD8^+^ T cells, and increase expression of granzymes and Fas, markers of cytotoxic T cell activity. However, high-level PD-1 expression, a marker of T cell activation and exhaustion [46], was also observed with treatment. This suggests a homeostatic response of strong CD8^+^ T cell activation followed by exhaustion. The ability of PD-1 antibody treatment to increase peripheral CD8^+^ T cell anti-tumour activity suggests that the PD-1 expression is associated with reduced CD8^+^ T cell function. CHK1i+LDHU promoted the increased expression of PD-L1 on tumour cells as reported previously [25] but there was also an as yet unidentified tumour associated immune cell population that strongly upregulated PD-L1. Increased expression of other immune checkpoint and exhaustion genes including TIM3, PD-L2, TIGIT and Lag3 was also observed although this varied between the two models investigated. The prominent accumulation of NKT cells was unexpected but its magnitude suggests that it is critical to the immune responses observed here. NKT cells have limited TCR variation but can either directly or indirectly trigger strong anti-tumour cytotoxic responses by regulating DC and macrophage and T cells responses [47,48]. The lack of effect of anti-PD-1 indicates that more dominant mechanisms maintained an immunosuppressive microenvironment. This is likely to be a consequence of the pronounced accumulation of immunosuppressive CD4^+^ Treg cells and FoxP3^+^NKT cells [40]. Both Treg and FoxP3^+^ NKT cells can be induced by TGFβ [40,49]. Although TGFβ expression was not changed with CHK1i+LDHU treatment, this cytokine is strongly expressed in these tumours suggesting that intra-tumourally recruited CD4^+^ T and NKT cells are converted into FoxP3^+^ regulatory cells by the high levels of TGFβ in the tumour microenvironment. The conversion of NKT cells to the FoxP3 expressing suppressive form is suggested by the commensurate decreases and increase in the respective populations at the later times after treatment. Whether TGFβ is produced by the tumour cells or by the highly enriched macrophage population that may represent up to 30% of the tumour mass [21] is at present unknown. The phenotype of macrophages in the untreated tumours is consistent with an alternatively activated M2-type, although both M1 and M2-type macrophages express high levels of TGFβ [50]. Other chemokines upregulated in either the tumour or infiltrating immune cells by CHK1i+LDHU, including CCL5 that directly recruits Tregs [51] or CXCL9, CXCL10, CXCL11 that recruit activated effector, T, NK and NKT cells, as well as Tregs [52,53], may also be responsible for the accumulation of Tregs. Interestingly, CXCL12 and its receptor CXCR4 that acts as a chemoattractant for Treg cells [52,53], were down-regulated with CHK1i combination treatment.

The large monocyte/macrophage tumour infiltrate is also altered by CHK1i+LDHU treatment, with downregulated expression of CD206 and CD163, markers of classically activated M2 macrophages, and increased expression of iNOS and Msr1, markers of alternatively activated M1 macrophages [50]. The anti-tumour activity of pro-inflammatory M1 macrophages adds innate responses to the adaptive responses. The DAMPs released by tumours cells treated with CHK1i+LDHU in vivo are also likely to contribute to the recruitment of inflammatory macrophages as suggested by the increased CCL2 expression [7,52]. We have previously shown that CHK1i+LDHU is effective at controlling human tumour growth as xenografts in immunocompromised mice, and that treatment recruited macrophages to the tumour [18]. Since immunocompromised nude mice do not have functional NKT cells [54], although all other innate immune responses are intact, this suggests that the innate immune systems are significant contributors to the anti-tumour response following CKH1i+LDHU treatment.

The increased expression of components of the complement pathway may also contribute to the immunosuppressive microenvironment. Complement system expression was shown to increase tumour growth and metastasis in a number of experimental mice cancer models, partly through increased immune suppression, either by directly reducing T cell recruitment or indirectly through increased recruitment of MDSC [55].

CHK1i+LDHU triggered a strong anti-tumour T cell response demonstrated by the very high proportion of tumour antigen reactive CD4^+^ and CD8^+^ T cells in the blood (and spleen; data not shown) of treated mice. Tumour infiltrating CD8^+^ T cells expressed high levels of PD-1 and CD4^+^ T cells expressed FoxP3 and CTLA4, suggested that immune suppression in the tumour microenvironment may be responsible for downregulation of cytotoxic T cell activity in the tumour. Strategies that block this local immunosuppression should reverse this effect. However, the apparent failure of PD-1 and CTLA4 inhibitors in our models was a surprise as both PD-1^+^CD8^+^ and CTLA4^+^ CD4^+^ T cells were increased with drug treatment. A similar CHK1i combination in SCLC was demonstrated to effectively combine with anti-PD-L1 to produce strong anti-tumour responses [25]. CHK1i+LDHU triggered a variable level of PD-L1 expression in vitro in the melanoma cell lines used here, that appeared weaker than reported for a similar CHK1i combination in lung cancer models [25], although a stronger increase in plasma membrane staining of PD-L1 was observed in the YUMMUV melanomas that escaped immune control in vivo. If PD-L1 expression only increased at latter times and is associated with immune escape, this would be different from the lung cancer study where drug treatment alone was sufficient to promote strong PD-L1. This suggested that either inhibition of a single immune checkpoint was insufficient to overcome the immune suppression, or that the choice of immune checkpoint inhibitor or timing of its application relative to CHK1i+LDHU treatment is critical for treatment success. It may be that inhibiting TGFβ signalling in the tumour microenvironment to arrest differentiation into Treg and FoxP3^+^ NKT cells could be a more effective strategy to reduce immunosuppression in melanomas.

## 5. Conclusions

We provided evidence that targeting replication stress using CHK1i+LDHU promotes immunogenic cell death and pro-inflammatory cytokine and chemokine expression in the tumour cells that recruit innate and adaptive immune cells to mediate a significant proportion of the therapeutic effect observed. However, either homeostatic counteractions or tumour cytokine expression generates an immunosuppressive tumour microenvironment that promotes Treg and immunosuppressive NKT cell differentiation and immunosuppressive myeloid cell recruitment, which reduce the efficacy of the combination. The ineffectiveness of anti-PD-1 despite the increased PD-1 on CD8^+^ T cells demonstrates the necessity of identifying immunotherapy that best blocks the immunosuppression in the tumour microenvironment. The ability to use acute treatment with CHK1i+LDHU to gain control of tumour growth, and at the same time essentially immunising the patient with their tumour has the potential to avoid the development of resistance to CKH1i as treatment can be discontinued once tumour growth is controlled.

## Figures and Tables

**Figure 1 cancers-13-03733-f001:**
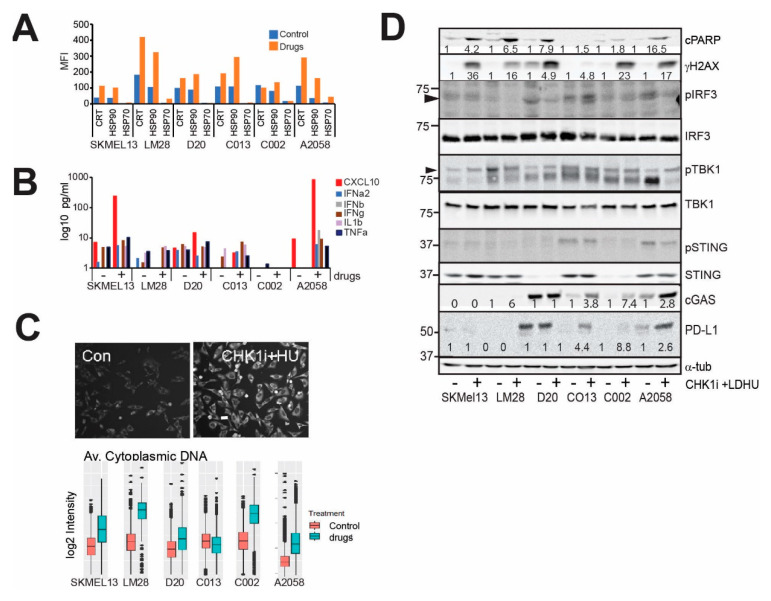
CHK1i+LDHU increases markers of ICD. (**A**) Human melanoma tumourspheres were treated with or without 0.2 μM SRA737 + 0.1 mM HU and harvested at 48 h for analysis of the cell surface expression of calreticulin (CRT), HSP70 and HSP90 on live cells. These data are representative of two independent experiments. (**B**) The indicated human melanoma tumoursphere lines were treated with or without 0.1 μM GDC-0575 + 0.1 mM HU for 48 h and the tissue culture supernatants were harvested and the cytokine levels measured. These data are representative of two independent experiments. (**C**) Immunostaining for cytoplasmic DNA in A2058 melanoma cells without and with treatment with 0.1 mM GDC-0575+0.1 mM HU (CHK1i+HU) for 48 h. Bar indicates 10 mm. The indicated melanoma cell lines were grown on plastic and treated as in B then fixed and stained for cytoplasmic DNA, and the level of staining was quantitated using high content imaging. The data are the mean and 95% CI from triplicate samples. (**D**) The indicated tumoursphere cultures were treated as in B and harvested at 24 h for immunoblotting of the indicated markers. The change in band intensity relative to the no-drug control for each cell line are shown. Where no changes are observed no quantitation is shown.

**Figure 2 cancers-13-03733-f002:**
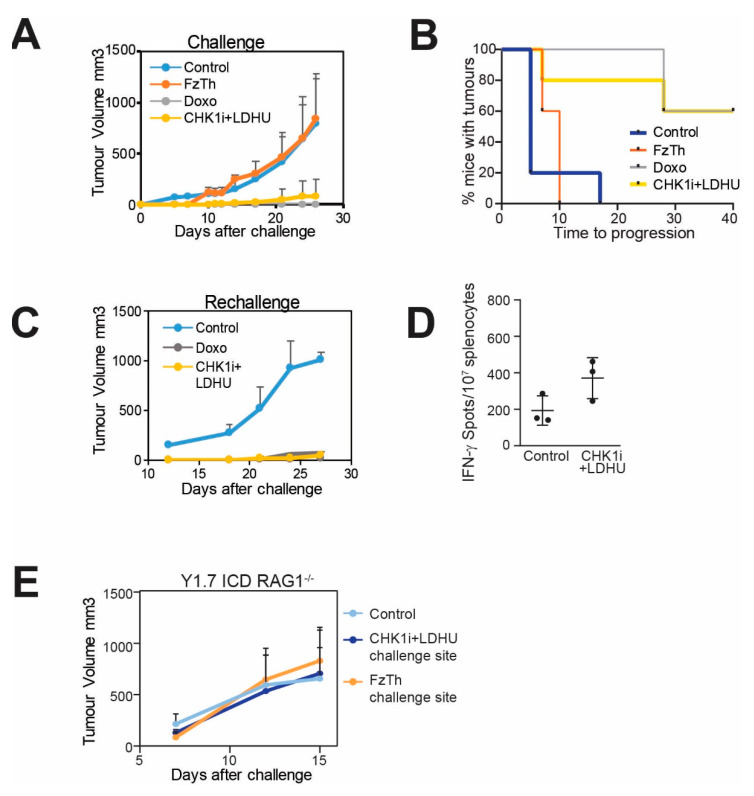
CHK1i+LDHU promotes ICD. (**A**) C57BL/6J mice were immunised with YUMMUV1.7 cells treated in vitro with either doxorubicin (Doxo) as a positive control for ICD or 0.2 mM SRA737+0.1 mM HU (CHK1i+LDHU) for 48 h. Freeze-thaw killed (FzTh) YUMMUV1.7 cells were used as a negative control for ICD. At 10 days after immunisation mice were rechallenged with live YUMMUV1.7 cells into the opposite flank and tumour growth followed. N = 5 mice each treatment. (**B**) Kaplan Meier graph of the same experiment as A, showing time to tumour progression of the rechallenge tumours. Progression was scored when tumours exceeded 50 mm^3^. (**C**) Additional control injected mice, and the mice that were protected by immunisation with either SRA737+LDHU or doxorubicin treated YUMMUV1.7 injected cells from A., were rechallenged with live YUMMUV1.7 cells 60 days after the original immunisation and tumour growth followed. (**D**) Splenocytes from either control or SRA737+LDHU treated YUMMUV1.7 immunised mice were challenged with 300cGy irradiated YUMMUV1.7 cells and an ELISpot assay of IFNγ was performed. (**E**) Rag1^−/−^ knockout mice were injected with the same cells as in panel A and tumour growth followed at the challenge site.

**Figure 3 cancers-13-03733-f003:**
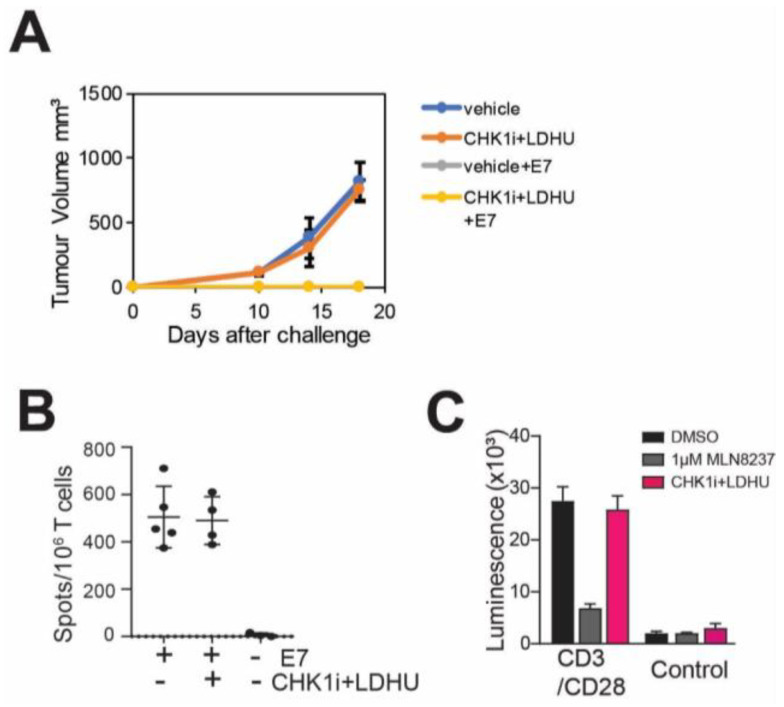
CHK1i+LDHU does not adversely affect a T cell-mediated immune response. (**A**) Immune competent C57BL/6J mice were treated with either vehicle or the normal three-week regimen of CHK1i+LDHU. In the second week of treatment, mice were immunised or not with MHC class I-restricted HPV E7 (RAHYNIVTF) peptide, and one week after completion of the drug treatment mice were challenged with live HPV E7 expressing syngeneic TC-1 tumour cells. Tumour growth was used as a measure of immune rejection. (**B**) Splenocytes isolated from the indicated mice from the experiment shown in A were stimulated with E7 peptide and an ELISpot assay was performed for the production of IFNγ. (**C**) Human donor-derived PBMC were treated with anti-CD3/CD28 beads to stimulate T cell proliferation for three days, then treated wither either vehicle (DMSO), 1 mM Alisertib or 0.2 mM SRA737+0.1 mM HU (CHK1i+LDHU) for three days, the drugs removed, and cells allowed to proliferate a further three days. Cell viability was assayed using CellTiter-Glo. The experiment was performed in triplicate and is representative of two independent experiments.

**Figure 4 cancers-13-03733-f004:**
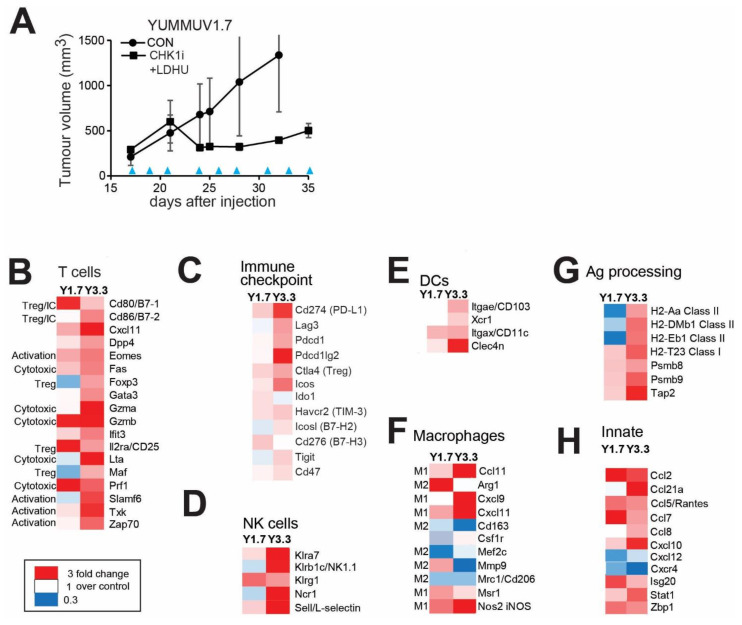
CHK1i+LDHU enhances adaptive and innate immune responses in vivo. (**A**) YUMMUV1.7 mouse melanoma line was established in immune-competent C57BL/6J mice (5 mice/group). When tumours had reached 200–300 mm^3^ they were treated three times/week every other day with 50 mg/kg SRA737 + 100 mg/kg HU by oral gavage for three weeks. Tumour volume was measured using callipers. The blue arrowheads indicate the treatment days. (**B–H**) Three tumours for each treatment were harvested 2 days after the final treatment and analysed using NanoString Mouse PanCancer Immune Profiling Panel. Heat maps of the changes in expression compared to vehicle-treated control of significantly altered genes are shown. The transcripts were partitioned into cell types and processes they are associated with.

**Figure 5 cancers-13-03733-f005:**
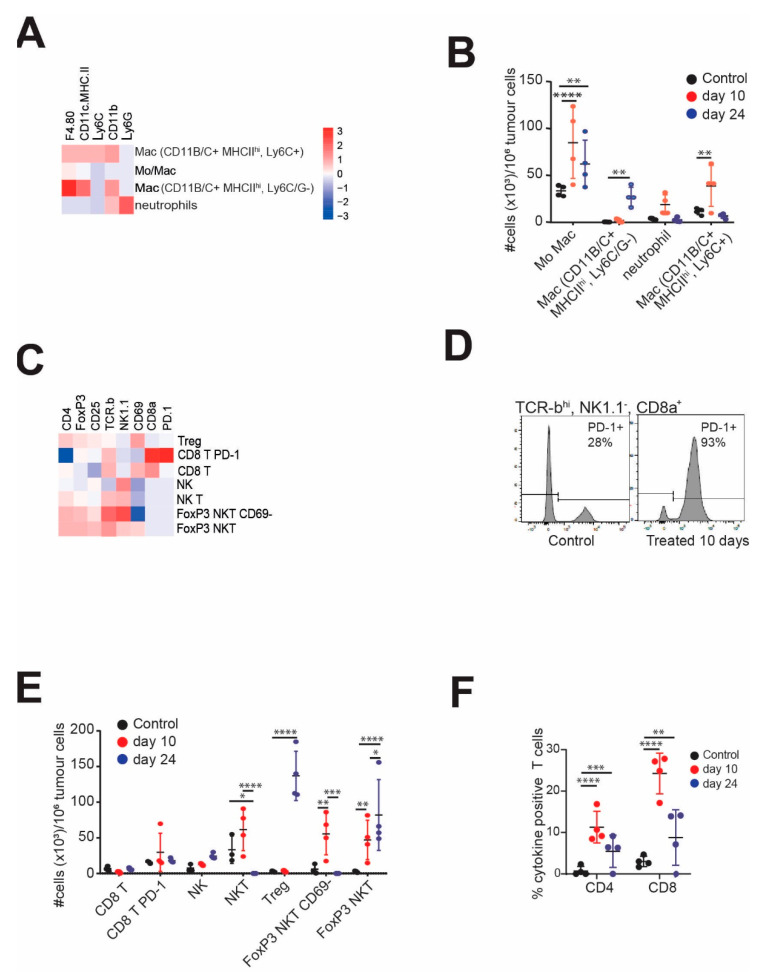
Myeloid and lymphoid cell profiling of YUMMUV1.7 tumours. YUMMUV1.7 tumour bearing mice were treated without or with CHK1i+LDHU for three weeks, three times/week. Untreated controls and treated mice were harvested on day 10 of treatment and treated mice three days after the final treatment (day 24). The tumours were dissociated and the CD45^+^ population analysed with either six marker panels of myeloid markers, or nine marker lymphoid sets using flow cytometry. Data were subjected to unsupervised clustering of the major clusters for each and quantitated for 4–5 mice for each time point. (**A**) Heatmap of marker staining intensity of major clusters for the myeloid markers and cell types they specify. (**B**) Absolute numbers of each population (per million tumour cells) for the replicate mice. (**C**) Heatmap of marker staining intensity of major clusters for the markers and cell types they specify. (**D**) Typical histogram of PD-1 staining of CD45^+^, TCRβ^hi^ CD8α^+^ NK1.1^−^ (CD8^+^ T) cells from control and 10 days treated YUMMUV1.7 tumour-bearing mice. (**E**) Absolute numbers of each population (per million tumour cells) for the replicate mice. (**F**) Percentage of tumour antigen-activated cytokine staining T Cells from the blood of the indicated mice at the indicated time points. * *p* < 0.05; **, *p* < 0.01, *** *p* < 0.001, **** *p* < 0.0001.

**Figure 6 cancers-13-03733-f006:**
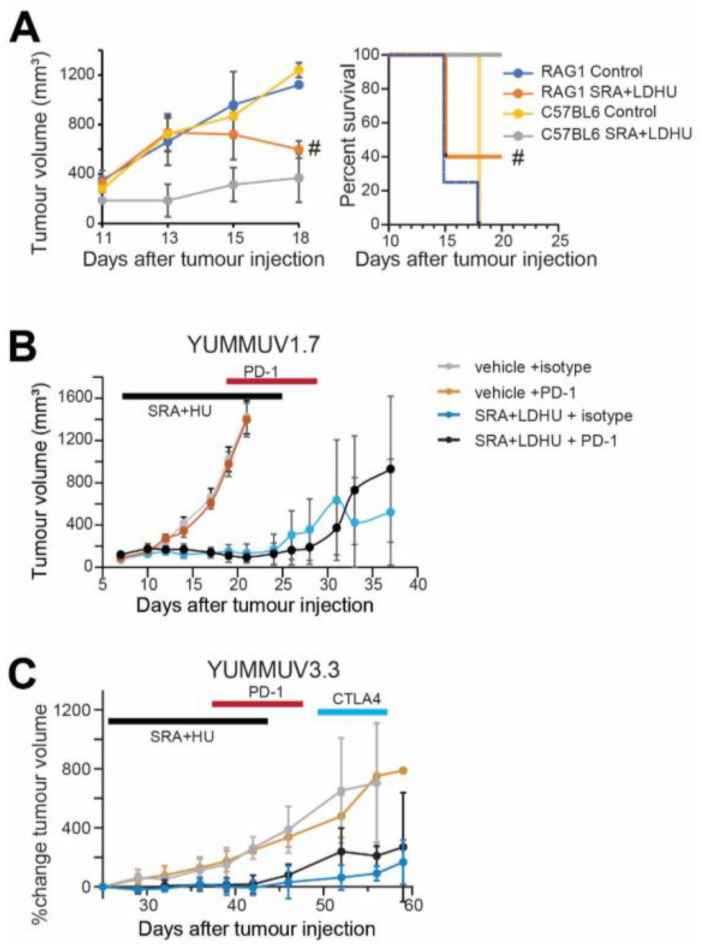
Tumour growth curves for CHK1i+LDHU ± immune checkpoint inhibitor-treated tumour-bearing mice. (**A**) YUMMUV1.7 tumours were established in Rag1^−/−^ (five mice/group) and immune-competent C57BL/6J (three mice/group) mice then treated from day 11 with SRA737+LDHU as previously. Tumour size was monitored. # indicates tumour size of the two remaining mice after three had reached ethical size limits (>1 cm^3^). Kaplan–Meier curve of the same experiment. (**B**) YUMMUV1.7 tumour bearing mice either without or with CHK1i+LDHU treatment, then treated with either isotype IgG or anti-PD-1. The data are for groups of 4–5 mice. The periods of treatment are indicated. (**C**) Similar data for YUMMUV3.3 mice expressed as percentage change from immediate pre-treatment volume. The individual mouse tumour growth is shown in Appendix A.

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
