# Peer review of "Targeting Replication Stress Using CHK1 Inhibitor Promotes Innate and NKT Cell Immune Responses and Tumour Regression"

_cancers, 2021, doi:10.3390/cancers13153733_

Round 1

Reviewer 1 Report

This is a well written article that is timely and relevant; especially with the current landscape of immune checkpoint inhibitor combinations and CHK1 combinations with other agents such as PARPi. I have some minor comments for the authors to address:

  1. Results point to the accumulation and major role for NKT regulatory cells in immunosuppression; yet authors chose not to deplete NKT cells to verify the suppressive role. At the minimum, suppressive assay would be beneficial.
  2. Please clarify why data with two separate CHK1 inhibitors were presented for separate experiments in the figure one. Please stay consistent and put the results with the different inhibitor in the supplementary. In vivo studies are consistent however.
  3. Authors should show quantifications for the western blots.
  4. Include high-magnification insets for the CD8 T cell histology.
  5. Without ELISpot or cytokine asssays with PD-1+ vs negative (and with and without PD-1 blockade in vitro), authors should not claim PD-1+ CD8 T cells are exhausted. Several studies have established that PD-1 expression alone does not define exhaustion. Was ELISpot or cytokine assay performed with TILs from 10 day time point as done with the peripheral cells?  If so was there a difference with or without PD-1 blockade?
  6. With regards to PD-1 and CTLA-4 blockade, authors may consider in future experiment to switch the timing of CTLA-4 vs PD-1 blockade as CTLA4 may have more pronounced effect in the priming phase.
  7. Any thoughts on combination with PARP inhibitors plus immune checkpoint blockade?

Author Response

  1. Results point to the accumulation and major role for NKT regulatory cells in immunosuppression; yet authors chose not to deplete NKT cells to verify the suppressive role. At the minimum, suppressive assay would be beneficial. 

We agree that this is an important experiment to perform and is a priority for our future research. However, the current manuscript describes the initial detailed investigation of the immune response triggered by treatment with CHK1i+LDHU, including the pro-inflammatory cytokine expression and immune cell infiltration and activation.  This includes myeloid and lymphoid compartments.  Both these are likely to be contributors to the immune response and importantly the immunosuppressive signals we see accumulate with treatment.  This manuscript forms the basis for future studies to define the relative contributions of each of these to identify more effective immune targets than the current immune checkpoint inhibitors, to enhance the immune response.

  1. Please clarify why data with two separate CHK1 inhibitors were presented for separate experiments in the figure one. Please stay consistent and put the results with the different inhibitor in the supplementary. In vivo studies are consistent however.

We have performed all of our previous studies using the Genetech drug and we wanted to demonstrate that the effects we have observed and reported here are not specific for one CHK1 inhibitor but are a general effect of this family of inhibitors.  We demonstrate in Supp Figure S1B that the two inhibitors have similar efficacy.  The action of this class of inhibitors is the focus of the current study rather than the efficacy of an individual molecule.  We have added a short section onto the discussion about this point. 

  1. Authors should show quantifications for the western blots.

We have added quantification as requested.

  1. Include high-magnification insets for the CD8 T cell histology.

We have increased the size of images in Supp Figure S5C to view the CD8 staining more clearly.

  1. Without ELISpot or cytokine assays with PD-1+ vs negative (and with and without PD-1 blockade in vitro), authors should not claim PD-1+ CD8 T cells are exhausted. Several studies have established that PD-1 expression alone does not define exhaustion. Was ELISpot or cytokine assay performed with TILs from 10 day time point as done with the peripheral cells?  If so was there a difference with or without PD-1 blockade?

We have not performed T cell assays on the TILs but on peripheral T cells and find that CD8+ T cell activity is higher in the PD-1 antibody treated mice than the isotype control mice (new Supp Figure S9B).  This suggests that PD-1 blockade is effective in maintaining CD8+ T cell activity although we agree it is not a definitive indicator of exhaustion.  We have modified the language in the Results and Discussion to reflect this. 

  1. With regards to PD-1 and CTLA-4 blockade, authors may consider in future experiment to switch the timing of CTLA-4 vs PD-1 blockade as CTLA4 may have more pronounced effect in the priming phase.

Thank you for the suggestion.  Our future research plan is to determine the contribution of CD8+ and NKT cells in the immune response and investigate means of block the accumulation of the immunosuppressive Treg and FoxP3+ NKT cells.  It is not clear whether the FoxP3+ NKT cells also express CTLA4.  This needs to be determined.  As we have stated in our Discussion, we also need to investigate whether targeting PD-L1 would be more effective as shown in the lung cancer models from the Byers’ lab.

  1. Any thoughts on combination with PARP inhibitors plus immune checkpoint blockade?

One of the reasons for our persisting with HU in combination with CHK1i is that we know we can titrate the well tolerated HU down to a point where it is not toxic to normal tissue, and the combination with CHK1i is minimally toxic, especially to the immune response.  We have not investigated the combination of CHK1i with PARPi, and although it has been reported in many cancer types to be highly effective, its effects on normal tissue or immune responses have not been reported.  If the role of PARPi in the combination is to simply increase replication stress by trapping PARP1/2 onto the DNA to acts as blocks for the replication fork, it may not be any more effective than HU.  

Reviewer 2 Report

  • A brief summary

Proctor and colleagues are showing in this manuscript that targeting the replication checkpoint kinase CHK1 (CHK1i) in combination with low dose of hydroxyurea (LDHU) block the tumor growth in mice models of melanoma. This effect of cancer elimination depends on inflammatory response and subsequent recruitment of immune cells that eliminate cancerous cells. However, CHKi + LDHU do not show additive effect with immune checkpoint blockade therapies (anti-PD1, anti-PD-L1).

  • Broad comments

These results are of broad interest to the all cancer research field. These date add to the recent progress in cancer therapies that try to exploit the inflammation induced by either replication defects or DNA damages to eliminate cancerous cells. Two points are however very surprising in the manuscript. (i) The inflammatory response here seems to be cGAS independent, but the underlying mechanism is eluded. (ii) CHK1i has been shown to be associated with side effects in patients, so that ATR inhibition which is less toxic in vitro and more tolerated by patients may be a more applicable to patients. ATRi is not assessed in this manuscript. I recommend minor revision of the paper before publication.

  • Specific comments

Minor comments.

  1. Whether the effect of CHK1i + LDHU is cGAS independent should be better verified and discussed with caution. Even if cGAS or STING levels are low or not detectable. The demonstration that the effect on tumor elimination can only be produced by depletion or knock-out of cGAS (or STING) in the cells and mice model used (see for instance the paper of Sen et al. Cancer Discovery 2019). Newly developed cGAS (RU.521) and STING (H151) inhibitors are also available to perform experiments. If the effect observed here is cGAS independent it should be explained to the readers that other DNA sensors or RNA sensors may induce interferon response upon CHK1i + LDHU. Here below I list a few references that should be discussed to justify the claim that the phenotypes are cGAS-STING independent.

Upon breakage of mitochondrial DNA, the RIG-I—MAVS pathway senses RNA to activate IFNs (Tigano et al., 2021). Even after gamma-irradiation, part of the inflammatory response depends on mitochondrial DNA damage (Tigano et al., Science 2021). For instance the inhibition of ATR in irradiated cells induces type I IFNs by the RNA sensing pathway RIG-I/MDA5 (Feng et al., EMBO J 2020). Other DNA sensors also exist and should be mentioned, TLR9/MYD88 (endosomes) and AIM2 (inflammosomes). To conclude some have shown that oncogenes (viral) have the capacity to inhibit cGAS or STING expression (Lau et al. Science 2015). Is it known whether in melanoma in which BRAF is mutated inhibition of cGAS or STING (activity or protein levels ; in 2/6 cell lines do not have STING or cGAS, in fig S2C) occurs as a way for cancerous cells to escape the immune system? Please, explain this issue and speculate.

  1. ATRi is not assessed. The paper would improve if the impact of ATRi would be assessed. In order to provide a broadening of the concepts suggested by the data. Then ATRi would then be more suitable to manage the potential side effects in patients. Last but not least I repeat that others have found that ATR inhibition potentiate type I IFNs by the RNA sensing pathway RIG-I/MDA5 (Feng et al., EMBO J 2020). What is the expected effect of the combination of ATRi and LDHU?

  1. Both PARPi and CHK1i have synergistic effects on tumor growth when administrated in combination with anti-PD-L1 immunotherapy in mice proficient for the cGAS-STING pathway. In STING- or cGAS-deficient mice, the combination of PARPi/antiPD-L1 or CHK1i/antiPD-L1 had no effect on tumor growth (Sen et al., Cancer Discovery 2019). These data show that under these circumstances, self-DNA sensing is essential to promote the immune rejection of cancer cells. Authors should speculate here about the discrepancies between their results and those of Sen et al. (the fact that immune checkpoint blockade synergize or not with the treatment).

  1. The mechanism of action proposed here is presumably more general and may apply to different types of cancers and different lines of treatments that destabilize genomic DNA (especially during DNA replication). Cite and discuss the work of Céline Gongora (Combès et al. Cancer Research 2019) showing that ATRi has anti-tumor effect in oxaliplatin resistant cancers through induction of inflammatory responses (again ATR). In addition PARP inhibitors (affecting DNA repair and fork stability) have been shown to produce similar effects in different cancer models (Chabanon et al. J Clin Invest 2019 ; Sen et al. Cancer Discovery 2019).

  1. P2 lines 84-85. The fact that CHK1 inhibition (+/- LDHU) leads to DNA damages, increase origin firing and cell death requires more adequate citations. Please refer to the work of Jiri Lukas (RPA exhaustion), Thomas Helleday (accumulation of DNA damages in CHK1i and increased origin firing) and of Michelle Debatisse (origin of DNA damages and consequences on DNA replication).

  1. The concept of selective killing tumors is very important in cancer research. However, CHK1 is essential for cell and organismal survival, HU will deplete the dNTP pool of all cycling cells. So that we can expect many side effect during the treatment or even later as a consequence of mutations and genome instability in healthy tissues. From authors data, is there a way to assess or monitor those short and long term side effects (blood analysis, resurgence of secondary tumors, others?)?

  1. Mechanistically, it is not explained why dNTP starvation potentiate CHK1i effects? Can authors explore the link, if any, between dNTP supply and CHK1 function (with appropriate citations)?

  1. Authors should incorporate the method for cytosolic DNA detection in the main text.

  1. Many references are not properly formatted, which has complicated the evaluation of the paper. For instance p5 line 216, refs to Sen and Li are not formatted.

Please check carefully the references as there are similar errors at several places in the manuscript.

  1. P7 lines 253-256. It should be stated in the text that experiments are performed in immunocompetent mice (so that authors do not need to go to materials and methods or figure legends to double check about this information).

  1. Figures S2 B and C may be integrated to the main figures.

Author Response

  • Broad comments

These results are of broad interest to the all cancer research field. These date add to the recent progress in cancer therapies that try to exploit the inflammation induced by either replication defects or DNA damages to eliminate cancerous cells. Two points are however very surprising in the manuscript. (i) The inflammatory response here seems to be cGAS independent, but the underlying mechanism is eluded. (ii) CHK1i has been shown to be associated with side effects in patients, so that ATR inhibition which is less toxic in vitro and more tolerated by patients may be a more applicable to patients. ATRi is not assessed in this manuscript. I recommend minor revision of the paper before publication.

In response to your first point, we do show that in the mouse model the cytokine expression we see is likely to be mediated by NFkB and we do show that RelA is phosphorylated in response to CHK1i+LDHU treatment in vitro (Supp Figure S6).  While we have not investigated this further at this point, the lack of not just STING and cGAS but absence of any obvious activation by phosphorylation of STING, TBK1 or IRF3 in any of the human melanoma cell lines or the mouse melanoma line supports this conclusion.  Furthermore, a very recent paper from Andrew Massey’s group has also shown that CHK1i fails to activate cGAS-STING.  We have added this reference to the Discussion.   The second point raised is about the preference for ATRi over CHK1i.  The reviewer is referencing CHK1i as single agents which have shown toxicity and very limited efficacy.  We have also shown that CHK1i as single agents have limited efficacy and this was our reason for using the combination of subclinical doses of CHK1i and hydroxyurea.  We have referenced this in our previous work (Oo et al 2019 ref18 in the original manuscript). We have also shown in the previous study and this study that the combination of CHK1i+LDHU used in this work has little obvious toxicity to rapidly proliferating tissue such as the colonic crypts and in blood white cell populations (Oo et al 2019, ref 18) and now in an adaptive immune response.  We also showed that manuscript that surprisingly ATRi did not synergise with LDHU.  Indeed we have published on this concept of targeting moderate replication stress as opposed to high level replication stress that is a target for ATRi (Nazareth et al 2019 ref 14). 

  • Specific comments

Minor comments.

  1. Whether the effect of CHK1i + LDHU is cGAS independent should be better verified and discussed with caution. Even if cGAS or STING levels are low or not detectable. The demonstration that the effect on tumor elimination can only be produced by depletion or knock-out of cGAS (or STING) in the cells and mice model used (see for instance the paper of Sen et al. Cancer Discovery 2019). Newly developed cGAS (RU.521) and STING (H151) inhibitors are also available to perform experiments. If the effect observed here is cGAS independent it should be explained to the readers that other DNA sensors or RNA sensors may induce interferon response upon CHK1i + LDHU. Here below I list a few references that should be discussed to justify the claim that the phenotypes are cGAS-STING independent.

Upon breakage of mitochondrial DNA, the RIG-I—MAVS pathway senses RNA to activate IFNs (Tigano et al., 2021). Even after gamma-irradiation, part of the inflammatory response depends on mitochondrial DNA damage (Tigano et al., Science 2021). For instance the inhibition of ATR in irradiated cells induces type I IFNs by the RNA sensing pathway RIG-I/MDA5 (Feng et al., EMBO J 2020). Other DNA sensors also exist and should be mentioned, TLR9/MYD88 (endosomes) and AIM2 (inflammosomes). To conclude some have shown that oncogenes (viral) have the capacity to inhibit cGAS or STING expression (Lau et al. Science 2015). Is it known whether in melanoma in which BRAF is mutated inhibition of cGAS or STING (activity or protein levels ; in 2/6 cell lines do not have STING or cGAS, in fig S2C) occurs as a way for cancerous cells to escape the immune system? Please, explain this issue and speculate.

Thank you for the suggestion.  We agree that the only definitive way of discounting cGAS-STING is to use deletion of these, however as stated above, the lack of effect of CHK1i on cGAS-STING has recently been have reported, and this reference is now included in the Discussion.  We also state in the Results section that many of pro-inflammatory cytokines we see expressed are NFkB targets and that the cytoplasmic DNA sensor ZBP1 that is strongly upregulated in response to CHK1i+LDHU in vivo can also activate NF-kB.  We have included an expanded discussion of cytoplasmic DNA sensing proteins and their potential contribution to the cell death and pro-inflammatory cytokine expression.  

  1. ATRi is not assessed. The paper would improve if the impact of ATRi would be assessed. In order to provide a broadening of the concepts suggested by the data. Then ATRi would then be more suitable to manage the potential side effects in patients. Last but not least I repeat that others have found that ATR inhibition potentiate type I IFNs by the RNA sensing pathway RIG-I/MDA5 (Feng et al., EMBO J 2020). What is the expected effect of the combination of ATRi and LDHU?

 As stated above, we have focussed on the subclinical CHK1i+LDHU rather than CHK1i as a single agent which has been shown to have limited clinical activity and toxicities.  We have shown in multiple assays that this combination has little normal tissue toxicity, even in highly proliferative tissue.   Importantly, we directly demonstrate that this combination does not adversely affect an adaptive immune response, a critical aspect of this work and of drugs expected to combine with immunotherapies.  We have reported in Oo et al 2019, that ATRi do not combine with LDHU.  We see increased cytoplasmic dsDNA so it is logical that our next line of investigation is other cytoplasmic DNA sensors, ZBP1 leading this investigation. 

  1. Both PARPi and CHK1i have synergistic effects on tumor growth when administrated in combination with anti-PD-L1 immunotherapy in mice proficient for the cGAS-STING pathway. In STING- or cGAS-deficient mice, the combination of PARPi/antiPD-L1 or CHK1i/antiPD-L1 had no effect on tumor growth (Sen et al., Cancer Discovery 2019). These data show that under these circumstances, self-DNA sensing is essential to promote the immune rejection of cancer cells. Authors should speculate here about the discrepancies between their results and those of Sen et al. (the fact that immune checkpoint blockade synergize or not with the treatment).

In our Discussion, we have suggested that targeting PD-L1 might be more effective, similar to the approach taken by the work from the Byers lab (Sen et al).  As cGAS-STING surprisingly appears to not be involved in the pro-inflammatory cytokine expression we have observed response to CHK1i+LDHU one of the aims of our future studies is to identify the mechanism controlling this expression.  We have added a section speculating on the difference we have observed from the work from the Byers lab. 

  1. The mechanism of action proposed here is presumably more general and may apply to different types of cancers and different lines of treatments that destabilize genomic DNA (especially during DNA replication). Cite and discuss the work of Céline Gongora (Combès et al. Cancer Research 2019) showing that ATRi has anti-tumor effect in oxaliplatin resistant cancers through induction of inflammatory responses (again ATR). In addition PARP inhibitors (affecting DNA repair and fork stability) have been shown to produce similar effects in different cancer models (Chabanon et al. J Clin Invest 2019 ; Sen et al. Cancer Discovery 2019).

It is not clear to us that the mechanism we are investigating with CHK1i+LDHU is the same as ATRi or PARPi. We have shown that ATRi does not synergise with LDHU, and we expected that PARPi are promoting high level replication stress which is a different mechanism from the moderate replication stress we are targeting in this study with CHK1i+LDHU (see Nazareth et al 2019 ref 14 for a discussion of this concept).   

  1. P2 lines 84-85. The fact that CHK1 inhibition (+/- LDHU) leads to DNA damages, increase origin firing and cell death requires more adequate citations. Please refer to the work of Jiri Lukas (RPA exhaustion), Thomas Helleday (accumulation of DNA damages in CHK1i and increased origin firing) and of Michelle Debatisse (origin of DNA damages and consequences on DNA replication).

Our previous work (Oo et al 2018 ref 11, Oo et al 2019 ref 18) both focus on the mechanism by which CHK1i -/+ LDHU function and work of many of the authors mentioned above have been cited in both papers.  This current manuscript is focused on the immune response triggered by this combination, and thus of Discussion is focussed on this aspect of the work as is appropriate.  

  1. The concept of selective killing tumors is very important in cancer research. However, CHK1 is essential for cell and organismal survival, HU will deplete the dNTP pool of all cycling cells. So that we can expect many side effect during the treatment or even later as a consequence of mutations and genome instability in healthy tissues. From authors data, is there a way to assess or monitor those short and long term side effects (blood analysis, resurgence of secondary tumors, others?)?

 As we have stated above, we have looked extensively at normal tissue responses in highly proliferative tissue such as blood and colonic crypts to our combination treatment in our previous paper (Oo et al 2019. Ref 18).  We reported there that the combination had minimal toxicity.  This has been mentioned in the Introduction of this manuscript. 

  1. Mechanistically, it is not explained why dNTP starvation potentiate CHK1i effects? Can authors explore the link, if any, between dNTP supply and CHK1 function (with appropriate citations)?

 This was investigated and discussed in our previous work (Oo et al 2018 ref 14).  As the current work is focussed on the immune response, we believe that having addressed this previously it is more appropriate to focus our Discussion on the immune responses being investigated here.  

  1. Authors should incorporate the method for cytosolic DNA detection in the main text.

 We have added this to the methods in the main text. 

  1. Many references are not properly formatted, which has complicated the evaluation of the paper. For instance p5 line 216, refs to Sen and Li are not formatted.

Please check carefully the references as there are similar errors at several places in the manuscript.

 Thanks you, we will check and correct these. 

  1. P7 lines 253-256. It should be stated in the text that experiments are performed in immunocompetent mice (so that authors do not need to go to materials and methods or figure legends to double check about this information).

 This has been added.

  1. Figures S2 B and C may be integrated to the main figures.

These have been added to the main section as requested.